# Effectiveness of Workplace Interventions for Improving Absenteeism, Productivity, and Work Ability of Employees: A Systematic Review and Meta-Analysis of Randomized Controlled Trials

**DOI:** 10.3390/ijerph17061901

**Published:** 2020-03-14

**Authors:** Lucia Tarro, Elisabet Llauradó, Gemma Ulldemolins, Pedro Hermoso, Rosa Solà

**Affiliations:** 1Universitat Rovira i Virgili, Facultat de Medicina i Ciències de la Salut, Functional Nutrition, Oxidation, and Cardiovascular Diseases Group (NFOC-Salut), Health Education and Promotion, 43201 Reus, Spain; lucia.tarro@urv.cat; 2Technological Unit of Nutrition and Health, EURECAT-Technology Centre of Catalonia, 43204 Reus, Spain; 3Activa Mutua, Mutua Colaboradora con la Seguridad Social, Tarragona, 43204 Catalonia, Spain; gemma.ulldemolins@activamutua.es (G.U.); phermoso@activamutua.es (P.H.); 4Hospital Universitari Sant Joan de Reus, Internal medicine department IISPV, Reus, 43204 Catalonia, Spain

**Keywords:** workplace intervention, productivity, performance, absenteeism, employees, work ability

## Abstract

To determine the effectiveness of workplace interventions and the most effective methodological design for the improvement of employee productivity, work ability, and absenteeism. A meta-analysis of randomized controlled trials (RCTs) of workplace interventions was conducted (PROSPERO, CRD42018094083). The PubMed, Scopus, PsycINFO, and Cochrane databases were searched. RCTs from 2000 to 2017 and with employees (18–65 years) were selected. Then, intervention characteristics and work-related outcomes data were extracted. A total of 47 RCTs were included in the systematic review, and 19 RCTs (11 absenteeism, 7 productivity, and 5 work ability) were included in the meta-analysis. The meta-analysis showed that the effectiveness of workplace interventions for absenteeism was −1.56 (95% CI, −2.67 to −0.44) and −2.65 (95% CI, −4.49 to −0.81) considering only moderate quality RCTs. In contrast, only a few studies of workplace interventions for productivity and work ability were included, which was insufficient for determining the effectiveness and best design for improving these work outcomes. The workplace is an interesting environment to reduce absenteeism, and individualized and counseling interventions with <10 sessions/total were the most effective workplace intervention methodological design for reducing the absenteeism of employees. Future high-quality RCTs that also consider health risks should be implemented to strengthen the results.

## 1. Introduction

Workplace interventions have emerged as a set of comprehensive health promotion and occupational health strategies implemented at the worksite to improve work-related outcomes [1] including improving productivity defined as fruitful working hours and performance [2]. Work-related outcomes are captured by different variables that reflect the employability of workers, with the most analyzed aspects being employee productivity, presenteeism, and absenteeism [3,4]. Employee productivity is assessed by the efficiency of employees (the time or resources required to develop a specific task) and their effectiveness (the objectives achieved and problems solved) [5]. In some studies, productivity represents the maximum individual potential performance at employees’ workplace [5]. Absenteeism is defined as the time that an employee is away from work due to illness or disability [2]. In contrast, presenteeism is defined as performing work at a lower level than usual during work time, with presenteeism problems not necessarily leading to absenteeism [2]. Consequently, both absenteeism and presenteeism could decrease the productivity of employees [2]. Additionally, another analyzed work-related aspect is work ability. Work ability has two definitions, namely, one definition for specific jobs that require special training, and one definition for jobs that most people can manage given a short period of practice [6].

The workplace is an excellent context for improving health outcomes because employees spend a substantial part of their days there [1]. Additionally, employers not only are responsible for providing a safe and hazard-free workplace but also have opportunities to promote and foster a healthy work environment [7]. As an extra benefit, the maintenance of a healthy lifestyle among employees reduces the direct and indirect costs [7]. Another interesting focus concerns the effects of workplace interventions on employee health lifestyle outcomes, such as physical activity, nutrition, weight reduction, or stress management [8,9], whereas it has not been defined how these lifestyle health aspects can contribute to improved work-related outcomes [10]. A meta-analysis of workplace interventions with the aim of improving the health of workers [11] concluded that their effectiveness is determined by the characteristics of the intervention. For example, young participants, sedentary employees, low participation at the start of the program, and weekly contact with participants are linked to greater effects from workplace interventions on the health of participants [11].

Evidence for the effectiveness of workplace interventions in improving the work-related outcomes of employees is still unknown. Although some literature on the effects of workplace interventions on work-related outcomes has been identified, they present inconsistencies in the results [12,13]. Although workplace interventions seem to offer some effectiveness in improving work ability outcomes [13] and decreasing presenteeism [12], other work-related outcomes such as employee productivity or performance still have not been evaluated.

Moreover, the characteristics of effective interventions for improving work-related outcomes have not yet been defined.

Thus, the aim of this systematic review and meta-analysis was to determine the effectiveness of workplace interventions for improving employees’ work-related outcomes such as productivity, work ability, and absenteeism; and the most effective methodological design for workplace interventions.

## 2. Materials and Methods

This systematic review and meta-analysis has a PROSPERO registration (number CRD42018094083).

### 2.1. Data Sources and Search Strategy

Four electronic databases, specifically, PubMed, Scopus, PsycINFO (American Psychological Association), and the Cochrane Central Register of Controlled Trials, were searched for randomized controlled trials (RCTs) published from January 2000 to December 2017. To screen the abstracts and full-text of the articles, the Covidence web-based software platform (http://www.covidence.org; Melbourne, Australia) was used to better manage and screen the high volume of identified articles. The search strategy was based on the Population, Intervention, Comparison, Outcome and Setting (PICOS) elements, which is a search tool used for organizing a framework to list terms by the main concepts in the search question [14]. PICOS elements were based on the following: population (“employees” and “18–65 years of age”); intervention (“worksite intervention” or “workplace intervention” or “work-site intervention” or “occupational”); comparison group (control group without any intervention or with a usual intervention); outcomes (“productivity” or “performance” or “presenteeism” or “absenteeism”); and setting (workplace).

### 2.2. Eligibility Criteria

The inclusion criteria of the interventions were as follows: randomized controlled trials (RCTs); workplace interventions aimed at improving employee productivity, performance, presenteeism or absenteeism, or a combination of these outcomes; published in English or Spanish; implemented and published between 2000 and 2017 (to focus on interventions conducted in a contemporary work environment); included employees aged 18–65 years; and included data about work-related outcomes as primary outcomes (employee productivity, work ability, performance, presenteeism, or absenteeism before and after the intervention and comparing an intervention group with a control group). The productivity, performance, and presenteeism parameters were assessed by using different tools although they express a concept similar to employee productivity; consequently, performance, presenteeism, and employee productivity were analyzed as a unique item: productivity.

The exclusion criteria were the absence of at least one inclusion criteria. Additionally, protocols or follow-up studies (in which the data that were collected after the end of the intervention were analyzed, whereas no control group was identified) were not considered. Finally, gray literature, correspondence letters, government statistical overviews, book chapters, dissertations, and conference abstracts were excluded from this systematic review and meta-analysis.

### 2.3. Definition of Outcomes

The work-related outcomes included in this systematic review and meta-analysis were absenteeism, productivity, and work ability. The absenteeism defined as time that employees are away from work due to illness or disability assessed as work hours or working days the employee was not at work [2]. The productivity was defined as the efficiency and effectiveness of employees doing their work, considering the time (presenteeism) or resources (performance) required for developing a specific task, and the problems solved for achieving their objectives [5]. For this reason, the productivity outcome included performance and presenteeism variables due to both variables expressing a similar concept although using different tools [2,5]. Lastly, work ability of employees was defined as the special training required for developing a specific task [6].

### 2.4. Data Collection and Analysis

#### 2.4.1. Selection of Studies

Two review authors (L.T. and E.L.) independently screened the title, abstract, keywords, and publication type of all articles identified with the search strategy described above from the electronic databases were screened. We resolved any uncertainties or disagreements by discussion. When the information in the title or abstract was insufficient to make a determination, the full manuscript text was read to determine whether the inclusion criteria were met. This selection process was performed by two reviewers (L.T. and E.L.). In cases of discrepancy in full-texts, a third reviewer (R.S.) was consulted.

#### 2.4.2. Data Extraction and Management

Data were extracted from each included study by two reviewers who worked independently (L.T. and E.L.). In cases of uncertainties or disagreements, a third reviewer (R.S.) was consulted for discussion. The data extraction sheets included the following items: name of publication, name of the intervention, authors, year of publication, year of intervention, city, country, study design, study objective, number of participants, age of participants, gender of participants, absence or presence of medical pathology in the included employees, number of participants in the intervention and control groups, type of intervention, duration of the intervention, description of the intervention, primary and secondary outcomes, and the tools used to measure the outcomes and results.

Study selection was performed according to the Preferred Reporting Items for Systematic Reviews and Meta-analysis (PRISMA; Appendix A), which is a guideline designed to prepare the systematic review and meta-analyses [15]. 

### 2.5. Quality Appraisal 

An analysis of the quality appraisal of the studies was included. The quality of the included studies was assessed as recommended in the Cochrane Handbook for Systematic Reviews of Interventions by using the standardized framework of the Quality Assessment Tool for Quantitative Studies developed by the Effective Public Health Practice Project [16]. This tool consists of eight items, namely, selection bias, study design, confounders, blinding, data collection methods, withdrawals and dropouts, intervention integrity, and analysis. For each item, the scores were summed according to the specific tool guidelines. This tool allows the categorization of each study’s methodological quality as weak (≥2 weak category ratings), moderate (1–3 strong category ratings and 1 weak rating), or strong (≥4 strong category ratings and no weak ratings) [16].

### 2.6. Data Processing for the Meta-Analysis

The RCT studies that met the inclusion criteria were selected for a systematic review, and if incorporated sufficiently, the data on absenteeism, productivity (performance and presenteeism), and work ability at baseline and at the end of the study were included in the meta-analysis. A study was included in more than one work-related outcome meta-analysis when it reported two or three of the aforementioned outcomes. The heterogeneity of the studies was evaluated by the I^2^ statistic. When the heterogeneity was over 75%, the studies were analyzed with randomized and non-fixed effects due to this indicate a considerable heterogeneity [17]. A *p*-value of <0.05 was considered statistically significant.

## 3. Results

The systematic search identified 7402 references, of which 125 articles were selected for a full-text manuscript review. After a full-text review, 47 RCT studies were found to meet the inclusion criteria and were included in the systematic review (Figure 1), and 19 RCTs were selected for the meta-analysis. 

### 3.1. RCT Studies’ Characteristics

The relevant information regarding the data extracted from the included studies is presented in Table 1, Table 2 and Table 3 and Appendix A.

Of the 47 RCTs that implemented workplace interventions to improve productivity (employee productivity, performance, and presenteeism), work ability, and absenteeism, 37 were RCTs (8100 participants), and 10 were cluster RCTs (2,456 participants); together, the 47 RCT studies included 10,556 participants.

Most of the RCT studies were conducted in the Netherlands (*n* = 10) [18,19,20,21,22,23,24,25,26,27], Denmark (*n* = 9) [28,29,30,31,32,33,34,35,36], the United States (*n* = 7) [37,38,39,40,41,42,43], and Japan (*n* = 5) [44,45,46,47,48], followed by Germany (*n* = 2) [49,50], Finland (*n* = 3) [51,52,53], and Australia (*n* = 3) [9,54,55]. Other represented countries included Brazil (*n* = 1) [56], Norway (*n* = 1) [57], Poland (*n* = 1) [58], Turkey (*n* = 1) [59], South Africa (*n* = 1) [60], Sweden (*n* = 1) [61], and Switzerland (*n* = 1) [62].

Of the 47 interventions, some focused on more than one outcome: 29 focused on improving absenteeism [18,19,21,22,23,24,26,27,28,29,31,33,34,35,38,40,42,49,50,51,52,53,54,55,56,57,58,59,60], 22 focused on increasing employee productivity [9,20,23,25,26,27,29,30,35,37,39,40,41,43,44,45,46,47,48,49,55,62], and 11 were dedicated to work ability [25,27,29,30,31,32,33,34,36,50,61].

Work ability was not defined as a keyword in the search strategy, but after the data extraction, a high number of the studies were found to consider work ability; thus, its results were incorporated in the present systematic review.

#### 3.1.1. Population Characteristics

Approximately half of the RCTs included in the systematic review, namely, 22 RCTs, focused on participants who worked in sedentary roles [9,18,21,22,25,26,30,35,37,38,39,40,42,43,48,49,50,51,56,59,61,62], such as office workers, administrators, or drivers, followed by 19 RCTs that focused on participants with non-sedentary jobs [23,24,27,28,29,31,32,33,34,36,41,44,45,46,47,54,55,57,60], such as health care workers, cleaning workers, etc.; five studies presented participants with sedentary and non-sedentary jobs [19,20,52,53,58]. Additionally, the subjects covered a large age range, from 18 to 67 years. In the majority of cases, the participants did not present any disease [9,22,23,25,26,27,29,30,31,32,33,34,35,37,38,40,41,42,45,46,47,48,52,54,56,57,58,60,61,62]. However, some of the articles included participants only if they had a specific disease or health disorder, such as musculoskeletal disorders [28,36,51,53,59], depression [18,20,44,50], a high risk of sickness absence [19,21,24], being overweight [39,55], stress [49], and cardiovascular risk [43].

#### 3.1.2. Intervention Characteristics

Five different types of interventions were identified. (1) Educational interventions (*n* = 17) were the most commonly used [9,19,20,25,29,32,37,41,42,43,44,45,46,49,52,54,62], followed by (2) multicomponent interventions (*n* = 11) [18,23,27,33,38,39,47,55,59,60,61], (3) counseling interventions (*n* = 9) [21,22,24,26,35,40,48,50,58], (4) physical activity interventions (*n* = 7) [28,30,31,34,36,56,57], and (5) organizational interventions (structural changes; *n* = 2) [51,53]. The duration of the interventions ranged from a one-time specific event to 36 months.

### 3.2. Work-Related Outcomes

Of the 47 RCTs, 32 were non-significant and 14 were significant in improving one of the work-related outcomes. Five RCTs reduced absenteeism [21,24,54,56,58], three RCTs increased work ability [32,36,61], five RCTs increased employee productivity (productivity, performance, or presenteeism) [41,45,46,47], one RCT increased productivity and absenteeism [55], and one RCT reduced presenteeism but did not effectively improve absenteeism [49].

Of the 14 interventions with significant results, nine focused on employees with a non-sedentary job [24,32,36,41,45,46,47,54,55], whereas only four addressed sedentary employees [21,49,56,61]. In contrast, of the 33 non-significant studies, 18 included sedentary employees [9,18,22,25,26,30,35,37,38,39,40,42,43,48,50,51,59,62], while 10 included employees with a non-sedentary job [23,27,28,29,31,33,34,44,57,60], and 4 included both types of workers [19,20,52,53].

#### 3.2.1. Absenteeism Parameters

In 22 studies, the effects of interventions for absenteeism were measured by the days or hours in which each participant was not present at his or her work and was evaluated; six of them observed a statistically significant decrease in absenteeism [21,24,54,55,56,58].

If we focused on effective interventions to reduce absenteeism, a 3-month multicomponent cognitive-behavioral intervention was implemented for the employees of postal and telecom services with sickness absences [24] based on four to five individual consultations with the occupational practitioner and a minimum of three contacts with company management. This intervention reduced the percentage of employees with sickness absences who returned to work within three months. Specifically, the time to return to work was significantly shorter in the intervention group (IG) than in the control group (CG; IG: 78% vs. CG: 63%; *p* = 0.02), and this reduction pattern was repeated for the duration of sickness leaves in days (IG: 49 vs. CG: 70; *p* < 0.01) [24]. Another multicomponent intervention, namely, a 3-month intervention based on weight loss directed toward the overweight and obese employees of an aluminum company, comprised an individual education session, a website to report one’s weight each week, the submission of daily food and physical activity diaries, a handbook with weight loss recommendations, and financial incentives [55]. This multicomponent weight loss intervention resulted in a significant decrease in the hours of absenteeism in the IG (−3.1 h (95% confidence interval (CI) −7.1 to 0.9)) compared to the CG (5.1 h (95% CI 0.5 to 9.6) *p* = 0.01) [55].

Moreover, a counseling intervention with a 12-month follow-up directed toward employees at a high risk of long-term sickness absences was based on an individual 30-minute structured consultation with an occupational practitioner [21]. As a result, the IG showed statistically lower sickness absences (days), with a mean ± standard deviation (SD) of 17.36 ± 28.25 days, than the CG (31.13 ± 55.47 days), *p* = 0.03 [22]. Another counseling intervention comprised of an 8-week mindfulness intervention directed toward the middle managers of insurance companies, banks, and advertising agencies to reduce their stress was based on eight meditation group trainings, one mindfulness session per day, and an individual follow-up session per participant. Ultimately, this intervention per participant achieved a greater reduction in absenteeism in the IG than in the CG (F(1,140) = 67.3, *p* < 0.001) [58].

Furthermore, a 24-week physical activity program with a 12-month follow-up directed toward bus drivers was implemented by fitness professionals [56] based on 3 or 4 training sessions per week of endurance exercise in the gymnasium with free weights and endurance exercise machines. The rate of absenteeism (mean ± SD) was higher in the CG than in the IG immediately after the intervention ended (CG: 0.69 ± 1.03 and IG: 0.17 ± 0.33; *p* < 0.05) and in the 12-month follow-up period (CG: 0.50 ± 0.46 and IG: 0.24 ± 0.32; *p* < 0.05) [56]. 

Finally, a 2-month mental health education training intervention based on mental health knowledge and communication for firefighter managers and a 6-month follow-up of all firefighters was associated with an absolute percentage point change of 0.28 (29% relative to baseline) hours in the CG and a change of −0.28 (−18% relative to baseline) hours in the IG (*p* = 0.049) [54].

Of the six significantly effective studies in the reduction of absenteeism, three stratified the target population either into employees at a high risk of sickness absences [21,24] or into overweight or obese employees [55]. In contrast, of the 22 non-significant studies related to the reduction of absenteeism, only eight stratified the population into employees with musculoskeletal disorders [28,51,53,59], employees with a high risk of sickness absences [19], stressed employees [49], and employees with depression [18,50]. Furthermore, the significant interventions presented 10 sessions as a maximum, whereas among the non-significant interventions, nine presented more than 10 sessions. 

#### 3.2.2. Work Ability Parameters

Eleven studies measured the effects of interventions for work ability parameters [25,27,29,30,31,32,33,34,36,50,61], nine of these assessed work ability by using the Workability Index Score (WAI), and only two used other questionnaires. High values of the WAI indicate a higher work ability. Three of the eleven studies reported significant results in the improvement of work ability [32,36,61].

One of the effective interventions was a 12-month multicomponent intervention applied to dentistry employees focused on 2 IGs and 1 CG [61]. One of the interventions was based on a mandatory 2 days/week of physical activity for employees inside of work hours, which did not show any improvement compared to the CG. The second intervention was based on the reduction of working hours from 40 h/week to 37.5 h/week, which improved the WAI values over the CG from 7.89 ± 1.94 to 8.09 ± 1.52 (*p* = 0.01). The CG did not receive any intervention [61]. 

One additional intervention that significantly improved work ability was based on a 10-week physical activity program applied to slaughterhouse workers with upper-limb chronic pain and work disabilities and included three sessions/week of 10 min of strength exercises for the shoulder, arm and hand muscles during work hours [36]. The IG showed an increase of 2.3 points in the WAI index (95% CI 0.9–3.7) compared to the CG (*p* < 0.05), which did not receive any intervention [36]. Another 10-week physical activity intervention applied to health care workers compared an IG who performed physical activity at work (5 × 10 min/week during work hours + 5 group-based physical activity motivational sessions) with a CG that performed physical exercises at home (5 × 10 min/week during leisure time) [32]. This intervention observed a significant improvement in the mean WAI index (95% CI) of 0.2 (−0.4, 0.9) points in the IG compared to a reduction of −0.9 (−1.6, −0.2) in the CG, which achieved a significant difference of 1.1 (0.3, 1.8) between the groups, *p* = 0.03 [32].

Of the three significantly effective interventions for work ability, two had a unique primary outcome [32,36]. However, the other non-significant interventions had another variable [25,29,33,34,50] or a large list of variables that included work ability as the primary outcome [27,30,31].

#### 3.2.3. Productivity, Performance, and Presenteeism Parameters

The effects on productivity, performance, and presenteeism parameters were assessed with different tools, which make it difficult to compare the results. However, we included these three outcomes together because of the similarities among them and because productivity encompasses both performance and presenteeism.

The effects on performance, productivity, and presenteeism were evaluated in 22 studies that treated productivity as a unique variable [9,20,23,25,26,27,29,30,35,37,39,40,41,43,44,45,46,47,48,49,55,62], and six of these studies observed a statistically significant improvement in productivity (productivity, performance, or presenteeism) [41,45,46,47,49,55].

A 3-month multicomponent weight loss intervention directed toward overweight and obese employees at an aluminum company (explained above in the absenteeism results) [55] significantly improved productivity as assessed by the Work Limitations Questionnaire (WLQ) when comparing the IG with the CG, −1.0 (−2.0 to 0.0) vs. 1.0 (−0.2 to 2.1), respectively, *p* = 0.01 [55]. Another 12-month multicomponent intervention that was directed toward all workers at an electronic dispositive company was based on three workshops to train leaders about stress, the work environment and organizational concepts, and from this, the participants developed action plans to improve their work environment [47]. Productivity was evaluated by the Health and Work Performance (HWP) questionnaire. Increased productivity in the IG compared to the CG was reported, with changes in HWP means of (±SD) 65.1 (±12.3) to 67.3 (±10.3) vs. 66.9 (±7.9) to 63.8 (±9.3), respectively (*p* = 0.048) [47].

A 16-week pilot intervention that focused on a tai chi intervention for nurses during work hours achieved a significant reduction in the WLQ score between the IGs (−3.1; ±1.2) and CGs (−0.8; ±1.4), *p* = 0.03 [41]. Additionally, a 7-week self-guided internet- and mobile-based stress intervention was applied to employees with perceived stress at an insurance company. Although it did not achieve significant effects for absenteeism, it achieved significant improvements in presenteeism (assessed as the number of “work cut back” days, which is reduced efficiency at work while feeling ill) with the Trimbos and Institute of Medical Technology Assessment Cost Questionnaire for Psychiatry (TiC–P-G) at a 6-month follow-up (but not at 7 weeks), and it reported a decrease in the IG compared to the CG of −3.3 (−4.6 to −2.0), *p* < 0.01 [49].

A 1-month educational intervention program directed toward the workers of an electric company was based on three group sessions (120 min/each) of cognitive-behavioral methods to improve performance and web-based exercises as homework [45]. Performance, as assessed by the HWP questionnaire, was increased in the IG compared to the CG (1.47 points (±0.30) vs. 0.69 points (±0.21), respectively; *p* = 0.04) [45]. Finally, another education program with a 3-month follow-up directed toward employees of a sake company was based on a 60-min lecture and 120-min practice session on the role of consultation and active listening [46]. Job performance assessed by the HWP questionnaire increased in the IG compared to the CG (F = 5.40, *p* = 0.029) [46].

### 3.3. Quality Appraisal and the Risk of Bias in the Included Studies

An initial analysis of the quality of the included studies showed that most of the studies had weak quality (≥2 weak category ratings; 28 of 47 total studies) [19,20,21,23,26,27,28,29,30,34,37,38,41,43,44,45,46,49,51,53,54,56,57,58,59,61,62], and the other studies presented moderate quality (18 of 47 studies) [18,22,24,25,31,32,33,35,36,39,40,42,47,48,50,52,55,60]. The most common quality flaw was blinding, which was not adequately performed in any of the studies. Selection bias, confounders, withdrawals, and dropouts were commonly avoided. Since most of the studies included in this systematic review had weak quality, the meta-analysis was performed by dividing the studies according to their quality (moderate or weak quality) to strengthen the generalization of the results obtained.

### 3.4. Meta-Analysis

After the systematic review was conducted, 19 of the 47 RCTs were selected for the meta-analysis [18,20,21,22,23,27,29,31,32,33,40,46,47,48,49,50,57,58,61]. Considering the quality of the 11 RCTs with absenteeism data that were included in the meta-analysis, five RCTs were of moderate quality [18,22,31,40,50] and six RCTs were of weak quality [21,27,29,49,57,58]. Additionally, for the seven RCTs with productivity data, three RCTs were of moderate quality [40,47,48] and four RCTs were of weak quality [20,23,27,46]. In addition, of the five RCTs that included work ability data, two RCTs were of moderate quality [32,33] and three RCTs were of weak quality [27,29,61]. Overall, nine moderate quality RCTs were included in the meta-analysis [18,22,31,32,33,40,47,50,63].

In the analysis of the effect size for absenteeism, of the 22 studies included in the systematic review, 11 were excluded for not incorporating absenteeism data (sickness absences in days) or the mean or SD values were missing. Thus, only 11 RCT publications that reported absenteeism data were included in the meta-analysis, including 2,195 participants in total [18,22,31,40,50]. Considering only the five moderate quality RCTs with absenteeism data, an effect size reduction of −2.65 days of sickness absences was observed (95% CI, −4.49 to −0.81; *p* < 0.001; Figure 2), which was confirmed in the six weak quality RCTs. However, both forest plots showed high heterogeneity (I^2^ statistic = 98%). 

Moreover, the RCTs were divided based on the methodological design of the intervention, such as face-to-face vs. virtual interventions, <10 sessions or ≥10 sessions per intervention, group vs. individual sessions and the type of intervention (counseling vs. non-counseling that included multicomponent interventions, physical activity and educational interventions). Nevertheless, high heterogeneity was observed in all cases (I^2^ statistic ≥ 97% depending on the methodological design). 

Face-to-face interventions were not effective for improving absenteeism, regardless of whether only moderate quality studies (absenteeism SD, −3.20; 95% CI, −9.87 to 3.47; *p* = 0.35) or moderate and weak quality studies were considered. However, virtual interventions were shown to be effective when considering moderate and weak quality studies together (absenteeism SD, −1.45; 95% CI, −2.40 to −0.50; *p* = 0.003). Nevertheless, when the weak quality studies were excluded from the meta-analysis, these interventions were not effective, although there was a trend (absenteeism SD, −1.72; 95% CI, −3.73 to −0.29; *p* = 0.09) (Figure 3).

On the other hand, considering the number of workplace intervention sessions, in the moderate quality studies, <10 intervention sessions demonstrated effectiveness (absenteeism SD, −3.99; 95% CI, −6.33 to −1.65; *p* < 0.001), and this result was confirmed after considering the weak quality studies (Figure 4).

Furthermore, comparing the group session and individual session interventions, individualized interventions were reported to be effective (absenteeism SD, −2.09; 95% CI, −3.06 to −1.13; *p* < 0.001), and this result was confirmed by considering the weak quality studies (Figure 5).

Moreover, when comparing the interventions based on the counseling and non-counseling interventions, counseling interventions were found to be effective for reducing absenteeism (absenteeism SD, −3.07; 95% CI, −4.69 to −1.45; *p* < 0.001; Figure 6); but this result was not confirmed by the weak quality studies. 

In the analysis of the productivity effect size, of the 22 studies in the systematic review that included productivity data, 15 RCTs were excluded for either not incorporating productivity data using the HPQ questionnaire and the mean or SD values were missing. Thus, only seven RCTs that reported productivity data were included in the meta-analysis, comprising 2,413 participants in total [20,23,27,40,46,47,48]. Considering only the three moderate quality RCTs with productivity data, there was a significant improvement in the productivity of employees, according to the effect size (0.33; 95% CI, 0.07 to 0.59; *p* = 0.01; Figure 7). However, this result was not confirmed by including the weak quality studies. Additionally, high heterogeneity was observed (I^2^ statistic = 98%). 

In the analysis of the effect size for work ability, of the 11 studies included in the systematic review, six were excluded for either not incorporating work ability data using the WAI index or the mean or SD values were missing. RCTs that used the entire WAI index or one of the items (item 1: current work ability compared to lifetime best, scored as 0 (completely unable to work) to 10 (work ability is at its best)) that appeared to be representative of the entire WAI index were incorporated in the meta-analysis. Thus, only five RCTs that reported work ability data were included in the meta-analysis, comprising 895 participants in total [27,29,32,33,64]. Considering only the two moderate quality RCTs with productivity data, there was a non-significant improvement of 0.48 in the effect size (95% CI, −0.47 to 1.43; *p* = 0.32) for employees’ work ability (Figure 8). However, this result was not confirmed when the weak quality studies were considered. Further, high heterogeneity was observed (I^2^ statistic = 98%). 

Due to the low number of RCTs on productivity and work ability outcomes, the effect of the methodological design of the interventions could not be determined. 

## 4. Discussion

This systematic review included 47 RCTs on workplace interventions, and 14 of these effectively improved one or more work-related outcomes. Specifically, six RCTs observed a decrease in absenteeism [21,24,54,55,56,58], three detected an improvement in work ability [32,36,61] and six noted an increase in employee productivity [41,45,46,47,49,55]. Focusing on the results of the meta-analysis of 19 RCTs [18,20,21,22,23,27,29,31,32,33,40,46,47,48,49,50,57,58,61], including 11 RCTs regarding absenteeism [18,21,22,27,29,31,40,49,50,57,58], 7 RCTs on productivity [20,23,27,40,46,47,48], and 5 RCTs concerning work ability [27,29,32,33,61], more than half of the studies were of weak quality. 

For this reason, we considered only the moderate quality RCT studies in the meta-analysis, and the workplace interventions that focused on reducing absenteeism were effective. Consequently, the workplace interventions effective for reducing absenteeism included the following methodological considerations: (a) counseling intervention design, (b) less than 10 sessions spread out across a maximum of 9 months, and (c) directed toward individuals instead of groups. However, the effectiveness of interventions focused on increasing employees’ work ability and productivity were ambiguous.

The interest in reducing absenteeism comes from the economic impact on governments and the company’s budget [65]. Although there is not a clear consensus about the effectiveness of workplace interventions for reducing absenteeism, the present meta-analysis contributes to increasing the evidence for designing and implementing effective interventions for reducing absenteeism. However, a systematic review published in 2013 about active workplace interventions to reduce sickness absences, which defined interventions as when the subject has an active role and when the goal is a behavioral change, concluded that the evidence available did not support active workplace interventions to reduce sickness absences [66]. Instead, another systematic review published in 2009 about the effectiveness of workplace interventions on work-related outcomes and health outcomes (musculoskeletal disorders, mental health problems, or other health conditions) concluded that for the musculoskeletal disorders subgroup, workplace interventions are effective in reducing sickness absences [67]. Finally, another systematic review published in 2004 concluded that the comprehensive treatment of low back pain via interventions had an effect on absenteeism, costs, and the prevention of new episodes of low back pain [68]. Thus, to our knowledge, the present meta-analysis is the first to provide data about of the effectiveness of RCTs developed at the workplace to reduce absenteeism. Interesting strategies to consider in the design of the interventions can be drawn from the current meta-analysis, because the most effective interventions focused on reducing absenteeism incorporated counseling and were based on individualized sessions of less than 10 sessions in total. Particularly, counseling interventions, which enable individuals to evaluate behavioral choices [69], can be defined as coaching, advising, mentoring, or motivational interviewing [70]. These types of interventions have demonstrated their effectiveness in reducing smoking and a tendency to increase physical activity in people with chronic obstructive pulmonary disease [71]. Another important characteristic of effective absenteeism reduction interventions was the individualization of the interventions instead of group sessions, a typical characteristic of counseling interventions. 

Moreover, although virtual interventions presented ambiguous results depending on whether only the moderate quality RCTs or both the moderate and weak quality RCTs were included, online interventions could be a promising strategy due to their increasing use for health-related issues [72]. Online or virtual interventions, instead of face-to-face interventions, present some advantages such as a reduced time and convenient location, the potential to access a larger target group, the anonymity of the participants, and reduced stigma depending on the focus on the project [73]. Web-based interventions have demonstrated benefits compared to face-to-face interventions in terms of improved knowledge or behavioral changes [73].

Regarding work ability, the present meta-analysis concluded that workplace interventions showed a non-significant increase of 0.54 (95% CI 0.03, 1.04), or 0.48 (95% CI −0.47, 1.43) when considering only the moderate quality RCTs. Similarly, our results are quite consistent with another recently published meta-analysis, which concluded that workplace interventions improve work ability by an increase of 0.12 (95% CI 0.03, 0.21) [13]. This meta-analysis showed a significant improvement, but the magnitude of the increase was smaller (0.12) than that obtained in the present meta-analysis (0.54). However, compared to the present systematic review that only focused on workplace interventions, the other systematic review has also incorporated interventions where only a component of the intervention occurs in the workplace, concluding that further RCT quality evidence is needed to arrive at a clear conclusion [13]. Considering that a decline in work ability with age is expected, with decreases of 0.5–0.7 points per year [74], an increase in work ability, independent of age, could be considered to be a gain. Furthermore, work ability is expected to decrease more in people with a non-sedentary profile such as installation and auxiliary workers, which requires a higher level of physical effort [74]. In addition, it is important to remark that in the studies incorporated in the present systematic review that demonstrated an increase in work ability, the success rate was even better in the employees who were categorized as non-sedentary. Therefore, specific and well-designed interventions are needed to improve the work ability of employees.

The 19 RCTs included in the meta-analysis of work-related outcomes were of weak or moderate quality, thus, future high-quality RCTs should be implemented to strengthen the results. Considering the tool used to categorize the quality of the included studies, any of them could be of strong quality. Of the eight items assessed, blinding was lacking or not reported in most of the studies, except one RCT, which reported single blinding [26]. Blinding in workplace interventions is extremely difficult [75] because most workplace interventions are social or behavioral interventions, where the employees are aware of the changes in their environment and in most cases, complete some self-report questionnaires [76,77]. Weak quality based on one item such as blinding means that the total score would not indicate strong quality. For this reason, all of the studies were, at maximum, of moderate quality. However, we decided not to exclude blinding at the risk of a biased questionnaire to assure transparency of the results described in the meta-analysis. However, the study design, i.e., RCT, was one of the inclusion criteria of the present meta-analysis and is also one aspect that makes the included studies strong.

Another aspect to consider in worksite interventions is employees’ socioeconomic factors, which can act as a confounder because employees of low socioeconomic status often perform high amounts of occupational physical activity [78]. However, which workplace interventions are effective for improving the work-related outcomes of employees with sedentary jobs is unknown.

Focusing on employee productivity outcomes, the current meta-analysis could not confirm the effectiveness of workplace interventions to increase employee productivity. The RCTs that provided health education training to intermediate managers achieved increased performance [46,47]. These results are in accordance with the RCTs aimed at promoting healthy lifestyles in school-based interventions, where adolescent leaders train close peers and achieve effective improvement in their healthy lifestyles [79]. Such peer-led methodologies are considered education between peers or close individuals with similar interests [80]. However, the two interventions included in the present review did not use a real peer-led methodology because only managers were trained [46,47], and for peer-led methodologies, a closer peer of the workers needed to be trained by leaders to drive beneficial changes. Furthermore, increases in employee productivity because of workplace interventions can be influenced by cultural factors. For example, three of the six RCTs that presented effective interventions for employee productivity outcomes were implemented in Japan, which suggests that the Japanese culture is an interacting factor because Japanese workers spend many hours working, are very obedient, and even work excessively [81,82].

There are limitations to the present meta-analysis. First, a remarkable methodological feature is the lack of information about other aspects of the workplace intervention methodologies. For example, information regarding whether the intervention was implemented during work hours or non-work hours was not provided in most of the RCTs. Moreover, there were no tools for assessing the principal outcomes and characteristics of the interventions, such as detailed descriptions of the intervention, if the intervention involved any homework, the duration of the sessions, if the intervention was face-to-face or online, if there was a person in charge of implementing the intervention, and other factors. We considered the mentioned methodological features to be interesting points for future recommendations for workplace interventions, while more quality studies are needed. Several of the RCTs included in this review had methodological problems (i.e., small sample sizes, an inadequate description of the interventions, weak quality, and other considerations). In addition, most of the studies had a short follow-up, which could represent a limitation because the long-term impact of these interventions on work-related outcomes and their sustainability are still unknown. On the other hand, the inclusion of workplace health risks and the type of instrument to monitor the health risks was lacking for many of the interventions included in the present systematic review (46 of the 47 RCTs included); this information should be included when determining the quality of publications about workplace interventions. Only one of the RCTs included in the present systematic review assessed health risks [43] using the Personal Wellness Profile [83]. However, 13 of the 47 articles included in the systematic review assessed psychological factors such as stress before and after the intervention with the aim of improving the management of stress in the workplace to improve work-related outcomes such as absenteeism, productivity, or work ability [9,19,37,40,41,43,47,48,49,55,57,59,60,61]. Psychological factors are important aspects to consider in workplace interventions because in 2015, 40% of the workers assessed in the National Health Service survey described feeling unwell due to stress, which was associated with an increase in absenteeism [84]. Future RCTs should abide to the following CONSORT criteria [85]: an adequate sample size, accurate descriptions of the intervention, the use of validated tools to evaluate the target group, and specific information about the target group of workers and the company. Second, the use of non-validated tools to assess the outcomes and the use of different tools to assess the same variable make the comparisons among studies difficult and do not allow for a meta-analysis. Third, some of the work-related outcomes were assessed by tools not designed to assess a specific outcome, i.e., productivity was assessed by a performance questionnaire. Fourth, all the analyses in the meta-analysis showed high heterogeneity, which is a limitation of the current study. Finally, although the study quality was not an inclusion criterion, the weakness of the majority of the included studies presents problems for the generalization of the results of this systematic review, which is the reason why we separated the results according to the quality of the studies.

In view of the results obtained in this meta-analysis, three RCTs described workplace interventions that were effective for reducing absenteeism, were of moderate quality, and incorporated methodological design considerations such as individualized and counseling interventions and <10 sessions/total, and all these parameters were considered important for improving the effectiveness of workplace interventions focused on absenteeism reduction [22,40,50]. Consequently, reproducing RCTs focused on absenteeism reduction using the highlighted methodological considerations [22,40,50] could further confirm the effectiveness of these types of interventions in the workplace. Moreover, these methodological considerations could be a tool for companies that need to reduce the absenteeism of their employees.

## 5. Conclusions

The present meta-analysis of RCT studies supported the workplace as an interesting environment to reduce absenteeism and determines some effective methodological characteristics for the interventions aimed to reduce absenteeism. Specifically, multi-component and counseling interventions, with virtual and individualized interventions and <10 sessions/total were the most effective methodologies to reduce absenteeism. In contrast, in productivity and work ability, few studies were included to specify the methodological considerations. Future high-quality RCTs that also consider health risks should be implemented to strengthen the results. 

## Figures and Tables

**Figure 1 ijerph-17-01901-f001:**
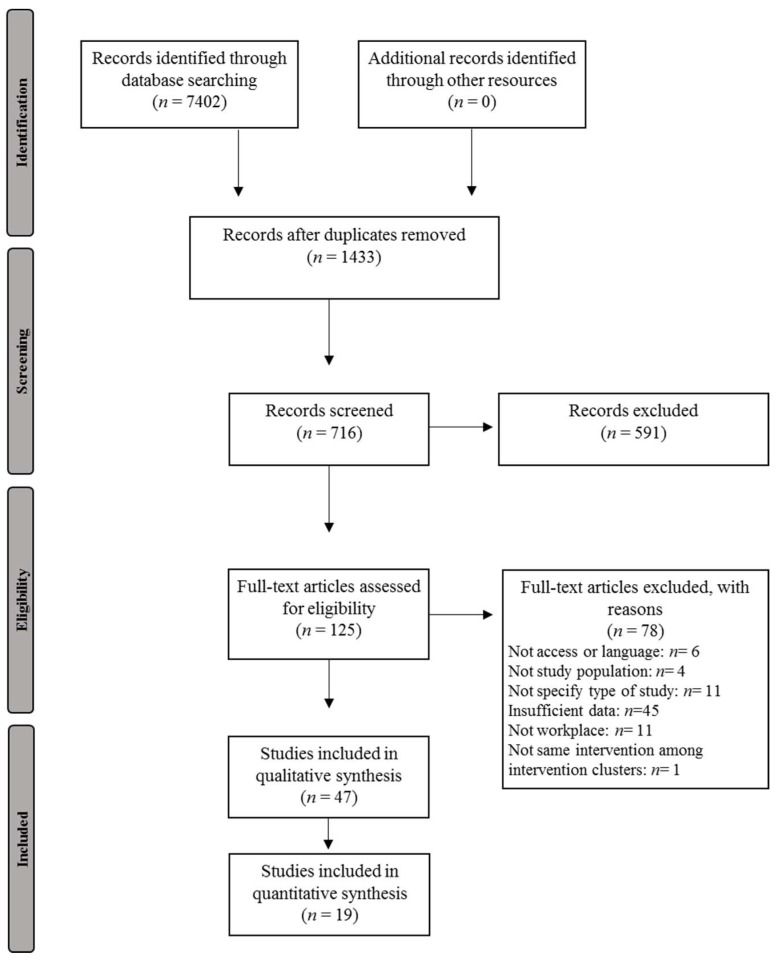
Flow diagram of included studies in the systematic review and meta-analysis.

**Figure 2 ijerph-17-01901-f002:**
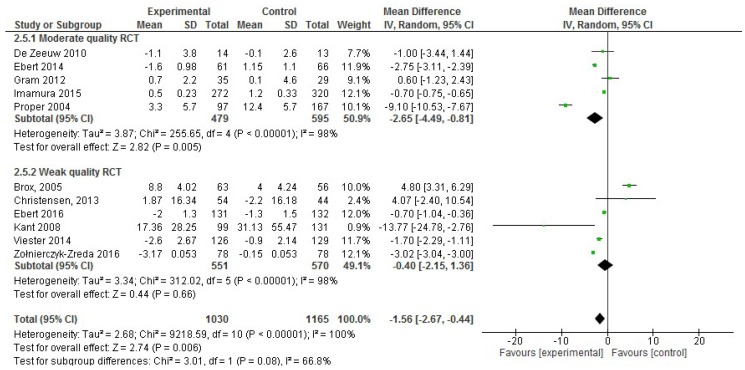
Forest plot of the effectiveness of workplace interventions for absenteeism reduction according to the quality of the studies.

**Figure 3 ijerph-17-01901-f003:**
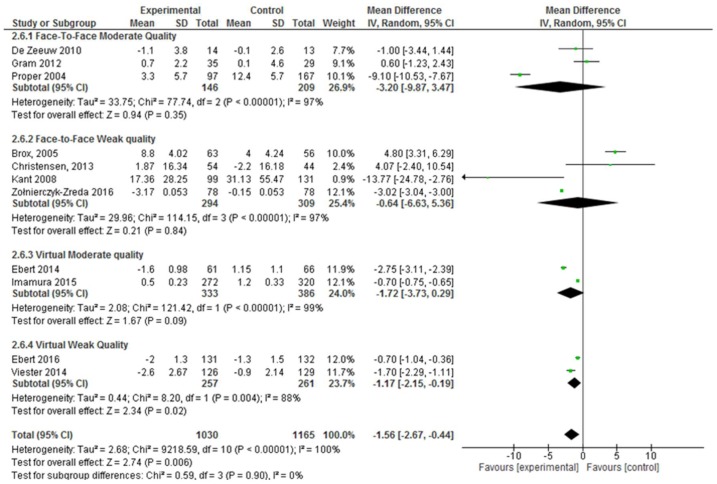
Forest plot of the effectiveness of face-to-face and virtual workplace interventions for reducing absenteeism according to the quality of the studies.

**Figure 4 ijerph-17-01901-f004:**
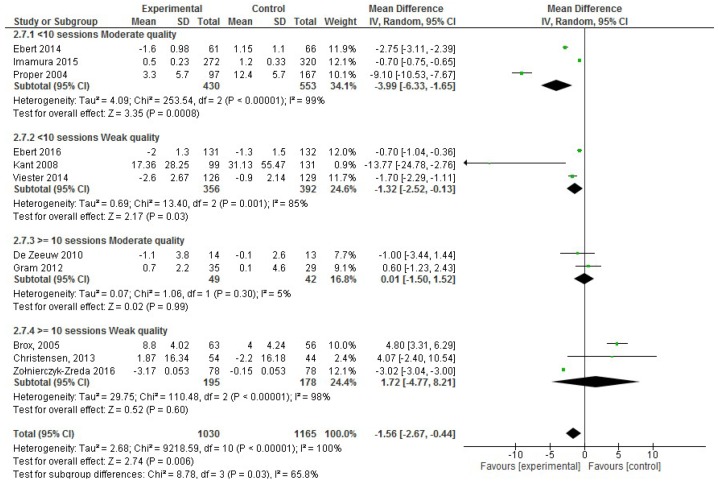
Forest plot for the effectiveness of workplace interventions for reducing absenteeism according to the number of intervention sessions and the quality of the studies.

**Figure 5 ijerph-17-01901-f005:**
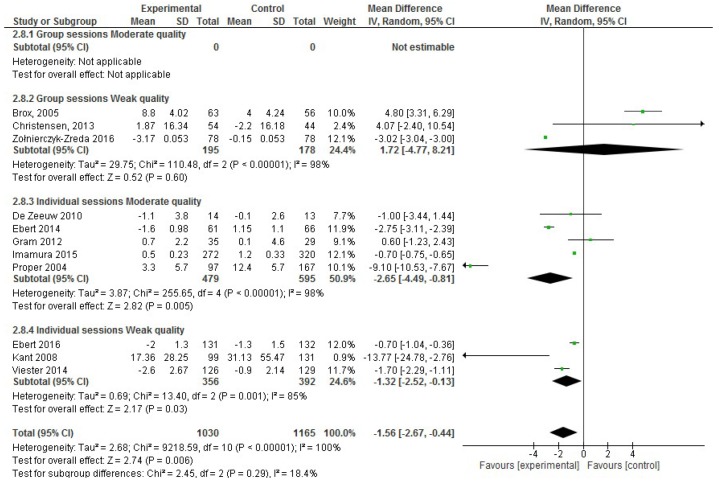
Forest plot of the effectiveness of workplace interventions for reducing absenteeism when considering whether the intervention included group or individual sessions and according to the quality of the studies.

**Figure 6 ijerph-17-01901-f006:**
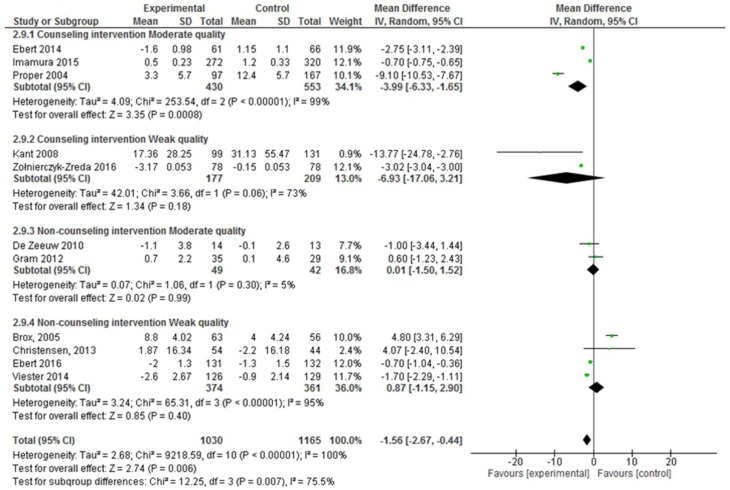
Forest plot of the effectiveness of counseling and non-counseling (multicomponent, physical activity and educational interventions) interventions for reducing absenteeism according to the quality of the studies.

**Figure 7 ijerph-17-01901-f007:**
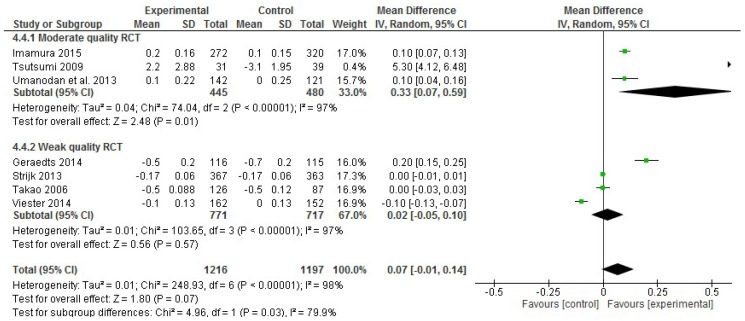
Forest plots of the effectiveness of workplace interventions for improving employees’ productivity according to the quality of the studies.

**Figure 8 ijerph-17-01901-f008:**
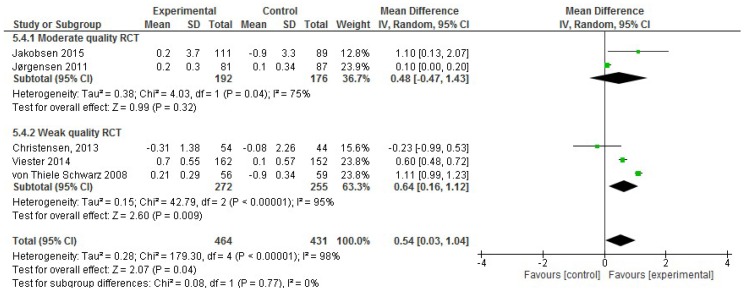
Forest plot of the effectiveness of workplace interventions for improving employees’ work ability according to the quality of the studies.

**Table 1 ijerph-17-01901-t001:** Intervention characteristics of the randomized controlled trials (RCTs) included in the systematic review focus on reducing absenteeism.

Ref	Population		Intervention Characteristics								Outcome	Results	Quality
	Type of Job	Stratification	Duration Interv + Follow-up	Type	Homework	Communication Channel	*N* of Sessions	Session Duration	Group or Individual	Time-Frame			
Arbogast, et al., 2016 [38]	Sed	No	13-month	MC	No	FtF	1	1 min	Ind	WH	ABS	Non-S	Weak
De Zeeuw et al, 2010 [18]	Sed	Yes	10-week	MC	No	FtF	20	40–50 min	Ind	WH	ABS	Non-S	Mod
Ebert et al, 2014 [50]	Sed	Yes	6-month	Cou	Yes	Onl	5	ns	Ind	ns	ABS + WAB	Non-S	Mod
Ebert et al, 2016 [49]	Sed	Yes	7-week + 6-month	Edu	Yes	Onl	7	45–60 min	Ind	ns	ABS + PRES	S (ABS: Non-S, PRES: S)	Weak
Esmaeilzadeh et al, 2014 [59]	Sed	Yes	6-month	MC	No	FtF	4	90 min	G + Ind	WH	ABS	Non-S	Weak
Imamura et al, 2015 [40]	Sed	No	6-week	Cou	Yes	Onl	6	30 min	Ind	ns	ABS + PERF	Non-S	Mod
Kant et al, 2008 [21]	Sed	Yes	specific time + 12-month	Cou	No	FtF	1	30 min	Ind	ns	ABS	S	Weak
Toppinen-Tanner et al, 2016 [52]	S/NoS	No	16-hour + 2 year	Edu	No	FtF	4	4 h	G	WH	ABS	Non-S	Mod
Proper et al, 2004 [22]	Sed	No	9-month	Cou	No	FtF	7	30 min	Ind	WH	ABS	Non-S	Mod
Reijonsaari et al, 2012 [35]	Sed	No	14-month	Cou	No	Tel + Onl	ns	ns	Ind	WH	ABS + PROD	Non-S	Mod
Shiri et al, 2011 [51]	Sed	Yes	Specific time + 12-month	Org	No	FtF	0	ns	Ind	WH	ABS	Non-S	Weak
Snetselaar et al, 2016 [42]	Sed	No	36-month	Edu	No	FtF	4	30 min	G	WH	ABS	Non-S	Mod
Van den Heuvel et al, 2003 [26]	Sed	No	8-week	Cou	No	Onl	40	Workday	Ind	WH	ABS + PROD	Non-S	Weak
Zavanela et al, 2012 [56]	Sed	No	24-week + 12-week	PA	No	FtF	36–48	ns	G	ns	ABS	S	Weak
Duijts 2008 [19]	Sed/NoS	Yes	6-month	Edu	No	FtF	7–9	1 h	Ind	ns	ABS	Non-S	Weak
Viikari-Juntura et al, 2012 [53]	S/NoS	Yes	12-month	Org	No	FtF	0	0	ns	WH	ABS	Non-S	Weak
Zołnierczyk-Zreda et al, 2016 [58]	Sed/NoS	No	8-week	Cou	Yes	FtF	10	180 min + 7 h	G + Ind	ns	ABS	S	Weak
Andersen et al, 2015 [28]	NoS	Yes	2-year	PA	No	FtF	30	50 min	G + Ind	WH	ABS	Non-S	Weak
Brox et al, 2005 [57]	NoS	No	1-week	PA	No	FtF	12	1 h	G	WH	ABS	Non-S	Weak
Christensen et al, 2013 [29]	NoS	No	13-month	Edu	Yes	FtF	48	1 h	G	WH	ABS + WAB + PERF, PROD, PRES	Non-S	Weak
Edries et al, 2013 [60]	NoS	No	6-week	MC	No	FtF	6	1 h	G	WH	ABS	Non-S	Mod
Gram et al, 2012 [31]	NoS	No	12-week	PA	No	FtF	12	12 min	Ind	WH	ABS + WAB	Non-S	Mod
Jørgensen et al, 2011 [33]	NoS	No	9-month	MC	No	FtF	30	20 min + 2 h (3-month) + 1 h (6-month)	G	WH	ABS + WAB	Non-S	Mod
Milligan-Saville et al, 2017 [54]	NoS	No	2-month + 6-month	Edu	No	FtF	1	4 h	G	ns	ABS	S	Weak
Morgan et al, 2012 [55]	NoS	Yes	3-month	MC	Yes	FtF + WB	1 FtF + WB	75 min	G	ns	ABS + PROD	S	Mod
Nurminen et al, 2002 [34]	NoS	No	8-month + 14-month	PA	No	FtF	26	60 min	G + Ind	WH	ABS+ WAB	Non-S	Weak
Strijk et al, 2013 [23]	NoS	No	6-month	MC	No	FtF	48 + 3 coach visits	45 min + 30 min (coach)	G + Ind	OutWH	ABS + PROD	Non-S	Weak
van der Klink et al, 2003 [24]	NoS	Yes	3-month + 9-month and 12-month	Cou	No	FtF	10	90 min	Ind	ns	ABS	S	Mod
Viester et al, 2015 [27]	NoS	No	4-month + 12-month	MC	No	FtF + Tel	3 FtF + 6 tel	60 min + 10–30 min (tel)	Ind	WH	ABS + WAB + PERF	Non-S	Weak

ABS: absenteeism; Cou: counseling; Edu: education; S: significant; Sed: sedentary; Sed/NoS: differ among employees; FtF: face-to-face; G: group; Ind: individual; MC: multicomponent; Mod: moderate; NoS: non-sedentary; ns: non-specified; Non-S: non-significant; Org: organizational; Onl: online; PA: physical activity; PERF: performance; PROD: productivity; PRES: presenteeism; Out WH: outside work hours; Tel: telephone; WAB: work ability; WB: web-based; WH: work hours. In the “Type of job” column, the studies were categorized among Sed, NoS or Sed/NoS. It divided studies depending the type of workers included. Sed (sedentary work type activity employees): workers of safe company; working teacher; insurance company workers; computer workers; information technology company employees; workers of bank; municipal organizations, governmental organizations and private enterprises; civil servant (white-collar workers); sedentary workers of different companies; officers; bus driver. NoS (non-sedentary work type activity employees): health care workers; manufacturing workers; construction workers; cleaning workers; firefighter; aluminum company; laundromat woman; postal and telecom services. Sed/NoS (sedentary and non-sedentary work type activity employees): health care and education sector workers; health workers and meat-processing industry and call centers; middle managers’ employees of insurance companies, banks and advertising agencies.

**Table 2 ijerph-17-01901-t002:** Intervention characteristics of the RCTs included in the systematic review focus on improving work ability.

Ref	Population		Intervention Characteristics								Outcome	Results	Quality
	Type of Job	Stratification	Duration Interv + Follow-up	Type	Homework	Communication Channel	*N* of Sessions	Session Duration	Group or Individual	Time-Frame			
Dalager et al, 2015 [30]	Sed	No	20-week	PA	No	FtF	20	1 h	G	WH	WAB + PROD	Non-S	Weak
Ebert et al, 2014 [50]	Sed	Yes	6-month	Cou	Yes	Onl	5	ns	Ind	ns	ABS + WAB	Non-S	Mod
VanBerkel et al, 2014 [25]	Sed	No	6-month + 12-month	Edu	Yes	FtF + Onl	8	90 min	G	OutWH	WAB + PERF	Non-S	Mod
von Thiele Schwarz et al, 2008 [61]	Sed	No	12-month	MC	No	FtF	ns	ns	ns	WH	WAB	S	Weak
Christensen et al, 2013 [29]	NoS	No	13-month	Edu	Yes	FtF	48	1 h	G	WH	ABS + WAB + PERF, PROD, PRES	Non-S	Weak
Gram et al, 2012 [31]	NoS	No	12-week	PA	No	FtF	12	12 min	Ind	WH	ABS + WAB	Non-S	Mod
Jakobsen et al, 2015 [32]	NoS	No	10-week	Edu	No	FtF	5 + PA	30–45 min	G	WH	WAB	S	Mod
Jørgensen et al, 2011 [33]	NoS	No	9-month	MC	No	FtF	30	20 min + 2 h (3-month) + 1 h (6-month)	G	WH	ABS + WAB	Non-S	Mod
Nurminen et al, 2002 [34]	NoS	No	8-month + 14-month	PA	No	FtF	26	60 min	G + Ind	WH	ABS+ WAB	Non-S	Weak
Sundstrup et al, 2014 [36]	NoS	Yes	10-week	PA	Yes	FtF	30	10 min	G	WH	WAB	S	Mod
Viester et al, 2015 [27]	NoS	No	4-month + 12-month	MC	No	FtF + Tel	3 FtF + 6 tel	60 min + 10–30 min (tel)	Ind	WH	ABS + WAB + PERF	Non-S	Weak

ABS: absenteeism; Cou: counseling; Edu: education; S: significant; Sed: sedentary; Sed/NoS: differ among employees; FtF: face-to-face; G: group; Ind: individual; MC: multicomponent; Mod: moderate; NoS: non-sedentary; ns: non-specified; Non-S: non-significant; Org: organizational; Onl: online; PA: physical activity; PERF: performance; PROD: productivity; PRES: presenteeism; Out WH: outside work hours; Tel: telephone; WAB: work ability; WB: web-based; WH: work hours. In the “Type of job” column, the studies were categorized among Sed, NoS or Sed/NoS. It divided studies depending the type of workers included. Sed (sedentary work type activity employees): office employees; working teacher; researchers; health workers (dentistry). NoS (non-sedentary work type activity employees): health workers; construction workers; cleaning workers; laundromat woman; slaughterhouse workers.

**Table 3 ijerph-17-01901-t003:** Intervention characteristics of the RCTs included in the systematic review focus on reducing productivity (presenteeism and performance).

Ref	Population		Intervention Characteristics								Outcome	Results	Quality
	Type of Job	Stratification	Duration Interv + Follow-up	Type	Homework	Communication Channel	*N* of Sessions	Session Duration	Group or Individual	Time-Frame			
Allexandre et al, 2016 [37]	Sed	No	8-week + 16-week	Edu	No	FtF + WB	8	1 h	G	WH	PROD	Non-S	Weak
Borness C, et al., 2013 [9]	Sed	No	16-week	Edu	No	Onl	48	20 min	Ind	WH	PROD	Non-S	Weak
Carr et al, 2016 [39]	Sed	Yes	16-week	MC	No	FtF + Onl	1 FtF + 3 emails/week	30 min	Ind	WH	PROD	Non-S	Mod
Terry et al, 2011 [43]	Sed	Yes	18-month	Edu	No	FtF + Tel	2 FtF + 11 tel or 1 FtF + 6 tel	ns	Ind	ns	PROD	Non-S	Weak
Donath L, et al., 2014 [62]	Sed	No	12-week	Edu	No	Onl	60	0	Ind	WH	PERF	Non-S	Weak
Umanodan et al, 2014 [48]	Sed	No	6-week	Cou	No	WB	6	30 min	Ind	WH	PERF	Non-S	Mod
Reijonsaari et al, 2012 [35]	Sed	No	14-month	Cou	No	Tel + Onl	ns	ns	Ind	WH	ABS + PROD	Non-S	Mod
Van den Heuvel et al, 2003 [26]	Sed	No	8-week	Cou	No	Onl	40	Workday	Ind	WH	ABS + PROD	Non-S	Weak
Ebert et al, 2016 [49]	Sed	Yes	7-week + 6-month	Edu	Yes	Onl	7	45–60 min	Ind	ns	ABS+ PRES	S (ABS: Non-S, PRES: S)	Weak
Imamura et al, 2015 [40]	Sed	No	6-week	Cou	Yes	Onl	6	30 min	Ind	ns	ABS + PERF	Non-S	Mod
Dalager et al, 2015 [30]	Sed	No	20-week	PA	No	FtF	20	1 h	G	WH	WAB + PROD	Non-S	Weak
Van Berkel et al, 2014 [25]	Sed	No	6-month + 12-month	Edu	Yes	FtF + Onl	8	90 min	G	OutWH	WAB + PERF	Non-S	Mod
Geraedts et al, 2014 [20]	Sed/NoS	Yes	8-week	Edu	Yes	Onl	6	ns	Ind	ns	PERF	Non-S	Weak
Furukawa et al, 2012 [44]	NoS	Yes	4-month	Edu	Yes	Tel + Onl	8	30–45 min	Ind	OutWH	PROD	Non-S	Weak
Kimura et al, 2015 [45]	NoS	No	1-month	Edu	Yes	FtF + WB	3	120 min	G	ns	PROD	S	Weak
Palumbo et al, 2012 [41]	NoS	No	16-week	Edu	Yes	FtF	5 workplace + 64 home	45 min (work) + 10 min (home)	G + Ind	WH	PROD	S	Weak
Takao et al, 2006 [46]	NoS	No	1-hour + 3-month	Edu	No	FtF	2	60 min + 120 min practice	G	ns	PERF	S	Weak
Tsutsumi et al, 2009 [47]	NoS	No	12-month	MC	No	FtF	3	ns	G	WH	PERF	S	Mod
Morgan et al, 2012 [55]	NoS	Yes	3-month	MC	Yes	FtF + WB	1 FtF + WB	75 min	G	ns	ABS + PROD	S	Mod
Strijk et al, 2013 [23]	NoS	No	6-month	MC	No	FtF	48 + 3 coach visits	45 min + 30 min (coach)	G + Ind	OutWH	ABS + PROD	Non-S	Weak
Viester et al, 2015 [27]	NoS	No	4-month + 12-month	MC	No	FtF + Tel	3 FtF + 6 tel	60 min + 10–30 min (tel)	Ind	WH	ABS + WAB + PERF	Non-S	Weak
Christensen et al, 2013 [29]	NoS	No	13-month	Edu	Yes	FtF	48	1 h	G	WH	ABS + WAB + PERF, PROD, PRES	Non-S	Weak

ABS: absenteeism; Cou: counseling; Edu: education; S: significant; Sed: sedentary; Sed/NoS: differ among employees; FtF: face-to-face; G: group; Ind: individual; MC: multicomponent; Mod: moderate; NoS: non-sedentary; ns: non-specified; Non-S: non-significant; Org: organizational; Onl: online; PA: physical activity; PERF: performance; PROD: productivity; PRES: presenteeism; Out WH: outside work hours; Tel: telephone; WAB: work ability; WB: web-based; WH: work hours. In the “Type of job” column, the studies were categorized among Sed, NoS or Sed/NoS. It divided studies depending the type of workers included. Sed (sedentary work type activity employees): collector (calling); white collar; office employees; health and airline workers; manufacturing workers; workers of safe company: insurance company workers; information technology; company employees; researchers. NoS (non-sedentary work type activity employees): workers of electric company; old nurses; workers of sake company; workers of electronic dispositive company; aluminum company; health workers; construction workers (blue collars). Sed/NoS (sedentary and non-sedentary work type activity employees): banking, research, security and university workers.

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
