# Peer review of "Effectiveness of Workplace Interventions for Improving Absenteeism, Productivity, and Work Ability of Employees: A Systematic Review and Meta-Analysis of Randomized Controlled Trials"

_ijerph, 2020, doi:10.3390/ijerph17061901_

Round 1

Reviewer 1 Report

This paper is of general interest. The quality of presentation is good. However, there are some suggestions for the authors.

I suggest changing the title to include the three outcomes (productivity, absenteeism,work ability).

Clearly define the description of objectives, including precise definitions of the variables and outcomes that are being evaluated.

An assessment and explicit acknowledgment of any researcher bias in the identification and selection of those studies shoul be integrated more precisely.

I propose to include a clearer description and evaluation of the degree of heterogeneity among the sample size of studies reviewed.

Reviewer 2 Report

This is a comprehensive, interesting, well-structured and well-written study. One problem is that most of the studies are with short follow-up even the one with three years. Another problem is that it is difficult to learn which of the study methods could be selected to prolonged studies to see the sustainability of the effect. Still there are methodological issues to learn from the study and it should be published with minor changes.

Please divide the tables with one table for each of the main outcomes: productivity, performance, presentation, absenteeism. Within each of these four categories please also rank the sedentary and non-sedentary job characteristics. This will make it possible to follow the abbreviations of the activities in the columns.

Another thing to be mentioned in the discussion is whether there has been a monitoring of the workplace health risks by using standardised psychosocial questionnaires (CoPSQ) or other type of monitoring instrument. This would be an important indicator if any described well known problems in the workplace is already known. If none of the studies are preceded from a monitoring of the occupational health risks, then just mention that none was found. And please mention in the conclusion that this would quality the studies.

Finally, as half of the studies or more are characterised to be of the week quality, it might be a good idea to remove the weak studies and make the table with explanation available in a link to the journal.
I think the authors should give their opinion of the usefulness of the first part of the article and the meta-analysis studies. Based on the meta-analysis, which of the studies would be authors recommend to be replicated in the future?

Round 2

Reviewer 1 Report

The manuscript was adapted accordingly to the suggestions and shows a significant improvement.